# GAL: Gradient Assisted Learning for Decentralized Multi-Organization Collaborations

**Enmao Diao**
Department of Electrical and Computer Engineering
Duke University
Durhm, NC 27705, USA
`enmao.diao@duke.edu`

**Jie Ding**
School of Statistics
University of Minnesota-Twin Cities
Minneapolis, MN 55455, USA
`dingj@umn.edu`

**Vahid Tarokh**
Department of Electrical and Computer Engineering
Duke University
Durhm, NC 27705, USA
`vahid.tarokh@duke.edu`

## Abstract

Collaborations among multiple organizations, such as financial institutions, medical centers, and retail markets in decentralized settings are crucial to providing improved service and performance. However, the underlying organizations may have little interest in sharing their local data, models, and objective functions. These requirements have created new challenges for multi-organization collaboration. In this work, we propose Gradient Assisted Learning (GAL), a new method for multiple organizations to assist each other in supervised learning tasks without sharing local data, models, and objective functions. In this framework, all participants collaboratively optimize the aggregate of local loss functions, and each participant autonomously builds its own model by iteratively fitting the gradients of the overarching objective function. We also provide asymptotic convergence analysis and practical case studies of GAL. Experimental studies demonstrate that GAL can achieve performance close to centralized learning when all data, models, and objective functions are fully disclosed. Our code is available here [1].

## 1 Introduction

One of the main challenges in harnessing the power of big data is the fusion of knowledge from numerous decentralized organizations that may have proprietary data, models, and objective functions. Due to various ethical and regulatory constraints, it may not be feasible for decentralized organizations to centralize their data and fully collaborate to learn a shared model. Thus, a large-scale autonomous decentralized learning method that can avoid data, models, and objective functions transparency may be of critical interest.

Cooperative learning may have various scientific and business applications [1]. As illustrated in Figure 1, a medical institute may be helped by multiple clinical laboratories and pharmaceutical entities to improve clinical treatment and facilitate scientific research [2, 3]. Financial organizations may collaborate with universities and insurance companies to predict loan default rates [4]. The organizations can match the correspondence with common identifiers, such as user identification associated with the registration of different online platforms, timestamps associated with different

---

[1]Resources related to Assisted Learning (AL) can be found at `http://www.assisted-learning.org`.

36th Conference on Neural Information Processing Systems (NeurIPS 2022).

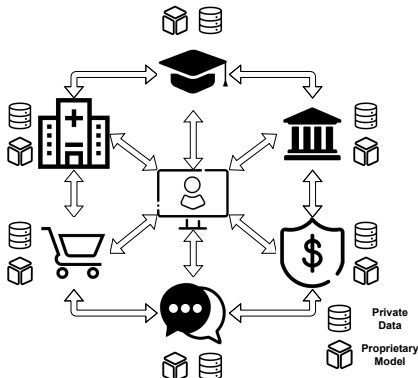

Figure 1: Decentralized organizations form a community of shared interest to provide better Machine-Learning-as-a-Service.

clinics and health providers, and geo-locations associated with map-related traffic and agricultural data. With the help of our framework, they can form a community of shared interest to provide better Machine-Learning-as-a-Service (MLaaS) [5, 6] without transmitting their local data, models, and objective functions.

The main idea of Gradient Assisted Learning (GAL) is outlined below. In the training stage, the organization to be assisted, denoted by Alice, will calculate a set of 'residuals' and broadcast these to other organizations. These residuals approximate the fastest direction of reducing the training loss in hindsight. Subsequently, other organizations will fit the residuals using their local data, models, and objective functions and send the fitted values back to Alice. Alice will then assign weights to each organization to best approximate the fastest direction of learning. Next, Alice will line-search for the optimal gradient assisted learning rate along the calculated direction of learning. The above procedure is repeated until Alice accomplishes sufficient learning. In the inference stage, other organizations will send their locally predicted values to Alice, who will then assemble them to generate the final prediction. We show that the number of assistance rounds needed to attain the centralized performance is often small (e.g., fewer than ten). This is appealing since GAL is primarily developed for large organizations with rich computation resources. A small number of interactions will reduce communication and networking costs. Our main contributions are summarized below.

- We propose a Gradient Assisted Learning (GAL) algorithm that is suitable for large-scale autonomous decentralized learning. It can effectively exploit task-relevant information preserved by vertically decentralized organizations. Our method enables simultaneous collaboration among organizations without sharing data, models, and objective functions.

- We provide asymptotic convergence analysis and practical case studies of GAL. For the case of vertically distributed data, GAL generalizes the classical Gradient Boosting algorithm.

- Our proposed method can significantly outperform learning baselines and achieve near-oracle performance on various benchmark datasets. Compared with existing works, GAL does not need frequent synchronization of organizations. It also significantly reduces the computation and communication overhead.

## 2   Related work

**Multimodal Data Fusion**  Vertically distributed data can be viewed as multimodal data with modalities provided in a distributed manner to different learners/organizations. Standard multimodal data fusion methods include the early, intermediate, and late data fusions [7,8]. These methods concatenate different modes of data at the input, intermediate representation, and final prediction levels. However, these data fusion methods in decentralized settings often require organizations to share the task labels to train their local models synchronously. In contrast, our method presented below only requires that organizations asynchronously fit some task-related statistics named pseudo-residuals to approximate the direction of reducing the global training loss in hindsight.

**Gradient Boosting** Our approach was inspired by Gradient Boosting [9,10], where weak learners are sequentially trained from the same dataset and aggregated into a strong learner. In our learning context, each organization uses side information from heterogeneous data sources to improve a particular learner's performance. Our method can be regarded as a generalization of Gradient Boosting to address decentralized learning with vertically distributed data.

**Federated Learning** Federated learning [11–16] is a popular distributed learning framework. Its main idea is to learn a joint model by averaging locally learned model parameters. It avoids the need for the transmission of local training data. Conceptually, the goal of Federated Learning is to exploit the resources of edge devices with communication efficiency. Vertical Federated Learning methods split sub-networks for local clients to jointly optimize a global model [17–22]. These methods can be viewed as federated learning with an intermediate data fusion method, and the central server will have access to the true labels. In order to converge, these methods typically require very frequent batchwise synchronization of backward gradients [22]. Frequent batchwise synchronization is critical for vertical federated learning because the local model at each client constitutes a part of the globally backpropable model, and one client's local update may not decrease the overall loss. In contrast, our proposed method trains multiple autonomous local models with pseudo-residuals, each contributing to a small portion of the overarching loss. Each round of updates will decrease the loss. Consequently, our method can achieve desirable performance with significantly fewer communication rounds without a global backpropable model.

**Assisted Learning** Assisted Learning (AL) [23] is a decentralized collaborative learning framework for organizations to improve their learning quality. In that context, neither the organization being assisted nor the assisting organizations share their local models and data. The original AL methodology applies to regression tasks. It is derived from a linear projection perspective, and its convergence to the oracle performance was theoretically justified for linear regression models with quadratic loss.

Inspired by Gradient Boosting, the proposed Gradient Assisted Learning (GAL) is a general method for multiple organizations to assist each other in supervised learning scenarios. Overall, AL and GAL share similar motivations and concepts but significantly differ from methodological and theoretical perspectives. More specifically, the novelties of GAL include: 1) generalization from regression loss to any differentiable loss for supervised learning, 2) allowing for local objective functions at each organization, 3) generalization from a sequential protocol between two organizations to parallel aggregation across multiple organizations, 4) introduction of deep model sharing for reducing computation and memory costs, 5) introduction of the assisted learning rate for fast convergence, and 6) theoretical analysis of GAL's convergence properties.

## 3 Gradient Assisted Learning

### 3.1 Notation

Suppose that there are $N$ data observations independently drawn from a joint distribution $p_{xy} = p_x p_{y|x}$, where $y \in \mathcal{Y}$ and $x \in \mathbb{R}^d$ respectively represent the task label and feature variables, and $d$ is the number of features. For regression tasks, we have $\mathcal{Y} = \mathbb{R}$. For $K$-class classification tasks, $\mathcal{Y} = \{e_1, \ldots, e_K\}$, where $e_k$ is the canonical vector representing the class $k$, $k = 1, \ldots, K$. Let $\mathbb{E}$ and $\mathbb{E}_N$ denote the expectation and empirical expectation, respectively. Thus, $\mathbb{E}_N g(y, x) \triangleq N^{-1} \sum_{i=1}^N g(y_i, x_i)$ for any measurable function $g$, where $(y_i, x_i)$ are i.i.d. observations from $p_{xy}$. Suppose that there are $M$ organizations. Each organization $m$ only holds $X_m$, a sub-vector of $X$ (illustrated in Figure 2). In general, we assume that the variables in $X_1, \ldots, X_M$ are disjoint in the presentation of our algorithm, although our method also allows the sharing of some variables. For example, one organization may observe demographic features for a mobile user cohort, and another organization holds health-related features of that cohort. Without loss of generality, we suppose that Alice, the organization to be assisted, has local data $x_1$ and task label $y_1$, while other $M - 1$ organizations are collaborators which assist Alice and have local data $x_2 \ldots x_M$. We use $1 : m$ and $1 : t$ to represent from the first to the $m^{\text{th}}$ organization and the $t^{\text{th}}$ assistance round, respectively.

### 3.2 Problem

For $m = 1, \ldots, M$, let $\{x_{i,m}\}_{i=1}^N$ denote the available data to the organization $m$. Thus, $N$ objects are simultaneously observed by $M$ organizations, each observing a subset of features from the $x \in \mathbb{R}^d$. Alice also has local task labels $\{y_{i,1}\}_{i=1}^N$ for training purposes.

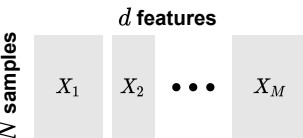

Figure 2: Illustration of organizations' vertically distributed data.

Let $\mathcal{F}_m$ and $L_m$ respectively denote supervised function class (such as generalized linear functions or neural networks) and the local objective function of organization $m$. We will assume that $L_m$ is differentiable. Without loss of generality, we assume that Alice denotes organization 1, who will be assisted. Without assistance from other organizations, Alice would learn a model that minimizes the following empirical risk,

$$F_{\text{Alone}} = \operatorname*{argmin}_{F_1 \in \mathcal{F}_1} \mathbb{E}_N L_1(y_1, F_1(x_1)). \tag{1}$$

Note that the above formulation only involves Alice's local data $x_1$ and local model (as represented by $F_1$ and $L_1$). In an oracle case, Alice would be able to operate on other organizations' data $x_2, \ldots, x_M$ as well. Recall that $x$ represents the ensemble of all the available data variables. In general, the ideal case for Alice is to minimize the following empirical risk,

$$F_{\text{Joint}} = \operatorname*{argmin}_{F \in \mathcal{F}} \mathbb{E}_N L_1(y_1, F(x)), \tag{2}$$

where $\mathcal{F}$ is a supervised function class defined with the space of $x$ as its domain.

In reality, Alice has no access to the complete data and model resources of other organizations. In this light, she shall be happy to crowdsource the learning task to other organizations to cooperatively build a model in hindsight, without the need to share any organization's local data or models. In the prediction stage, Alice can collect the information needed to form a final prediction to hopefully achieve a performance that significantly improves over her single-organization performance. To this end, we will develop a solution for Alice to achieve such a goal.

We will include detailed derivations and discussions of our solution in Subsection 3.3. For readability, we summarize notations that will be frequently used in the exposition below. Our method will require Alice to occasionally send continuous-valued vectors $r_1 = [r_{i,1}]_{i=1}^N \in \mathbb{R}^{N \times K}$ to each organization $m$ at each communication round. These residuals, to be elaborated in the next subsection, approximate the fastest direction of reducing the training loss in hindsight, namely a sample version of $\partial L_1(y_1, F(x))/\partial F(x)$ given Alice's estimation of $F$ at a particular time (round). Upon the input of these residual vectors, the organization $m$ will locally learn a supervised function $f_m$ that maps from its feature space to the residual space. With a slight abuse of notation, we also refer to $f_m$ as the learned model. To this end, the organization $m$ will perform the empirical risk minimization

$$f_m = \operatorname*{argmin}_{f \in \mathcal{F}_m} \mathbb{E}_N \ell_m(r_1, f(x_m)) = \operatorname*{argmin}_{f \in \mathcal{F}_m} \frac{1}{N} \sum_{i=1}^N \ell_m\left(r_{i,1}, f(x_{i,m})\right) \tag{3}$$

to obtain a locally trained model $f_m$. Here, $\mathcal{F}_m$ and $\ell_m$ respectively denote the supervised function class and loss function of the organization $m$. We note that $\ell_m$ are local regression loss functions for fitting the pseudo-residual $r_1$ and may not necessarily be the same as $L_1$ for fitting true labels. For example, $L_1$ may be the cross-entropy loss for classification of label $y$, while $\ell_{1:M}$ could be the squared loss for regression of the response $r_1$. The above local training (optimization) is often performed using the stochastic gradient descent (SGD) algorithm. In our assisted learning context, $L_1, \mathcal{F}_m$, and $\ell_m$ are proprietary local resources that cannot be shared across organizations.

## 3.3 The GAL Algorithm

We first introduce the derivation of the GAL algorithm from a functional gradient descent perspective. Then, we cast the algorithm into pseudocode and discuss each step. Consider the unrealistic case that Alice has all the data $x$ needed for a centralized supervised function $F : x \mapsto F(x)$. Recall that the goal of Alice is to minimize the population loss $\mathbb{E}_{p_{x,y}} L_1(y, F(x))$ over a data distribution $p_{x,y}$. If

---

**Algorithm 1** GAL: Gradient Assisted Learning (from the perspective of the service receiver, Alice)

---

**Input:** $M$ decentralized organizations, each holding data $\{x_{i,m}\}_{i=1}^N$ (local) corresponding to $N$ objects, the task label $\{y_{i,1}\}_{i=1}^N$ initially held by the service receiver (Alice local), model class $\mathcal{F}_m$ (local), gradient assistance weights $w$ (Alice local), assistance rate $\eta$ (Alice local), overarching loss function $L_1$ (Alice local), regression loss function $\ell_m$ to fit pseudo-residual (local), assistance rounds $T$.

**Learning Stage:**
> **Intialization:**
>> Let $t = 0$, and initialize $F^0(x) = \mathbb{E}_N(y_1)$
>
> **for** assistance round $t$ from 1 to $T$ **do**
>> Compute pseudo-residual
>> $$r_1^t = -\left[\frac{\partial L_1\big(y_1, F^{t-1}(x)\big)}{\partial F^{t-1}(x)}\right]$$
>> Broadcast pseudo-residual $r_1^t$ to other organizations
>> **for** organization $m$ from 1 to $M$ *in parallel* **do**
>>> $f_m^t = \arg\min_{f_m \in \mathcal{F}_m} \mathbb{E}_N \ell_m\left(r_1^t, f_m(x_m)\right)$
>>
>> **end**
>> Gather predictions $f_m^t(x_m)$, $m = 1, \ldots M$, from all the organizations
>> Optimize the gradient assistance weights
>> $$\hat{w}^t = \arg\min_{w \in P_M} \mathbb{E}_N \ell_1\left(r_1^t, \sum_{m=1}^M w_m f_m^t(x_m)\right)$$
>> Line-search for the gradient assisted learning rate
>> $$\hat{\eta}^t = \arg\min_{\eta \in \mathbb{R}} \mathbb{E}_N L_1\left(y_1, F^{t-1}(x) + \eta \sum_{m=1}^M \hat{w}_m^t f_m^t(x_m)\right)$$
>> $$F^t(x) = F^{t-1}(x) + \hat{\eta}^t \sum_{m=1}^M \hat{w}_m^t f_m^t(x_m)$$
>
> **end**

**Prediction Stage:**
> For each data observation $x^*$, of which $x_m^*$ is held by organization $m$:
> Gather predictions $f_m^t(x_m^*)$, $t = 1, \ldots, T$ from each organization $m$, $m = 1, \ldots, M$
> Predict with
> $$F^T(x^*) \triangleq F^0(x_1^*) + \sum_{t=1}^T \hat{\eta}^t \sum_{m=1}^M \hat{w}_m^t f_m^t(x_m^*)$$

---

$p_{x,y}$ is known, starting with an initial guess $F^0(x)$, Alice would have performed a gradient descent step in the form of

$$F^1 \leftarrow F^0 - \eta \cdot \frac{\partial}{\partial F} \mathbb{E}_{p_{x,y}} L_1(y, F(x)) \mid_{F=F^0} = F^0 - \eta \cdot \mathbb{E}_{p_{x,y}} \frac{\partial}{\partial F} L_1(y, F(x)) \mid_{F=F^0}, \quad (4)$$

where the equality holds under the standard regularity conditions of exchanging integration and differentiation. Note that the second term in (4) is a function on $\mathbb{R}^d$. However, because Alice only has access to her own data $x_1$, the expectation $\mathbb{E}_{p_{x,y}}$ cannot be realistically evaluated. Therefore, we need to approximate it with functions in a pre-specified function set. In other words, we will find $f$ from $\mathcal{F}_M$ that 'best' approximates $\mathbb{E}_{p_{x,y}} \frac{\partial}{\partial F} L_1(y, F(x))$. We will show that this is actionable without requiring the organizations to share proprietary data, models, and objective functions.

Recall that $\mathcal{F}_m$ is the function set locally used by the organization $m$, and $x_m$ is a correspondingly observed portion of $x$. The function class that we propose to approximate the second term in (4) is

$$\mathcal{F}_M = \left\{ f : x \mapsto \sum_{m=1}^M w_m f_m(x_m), \forall f_m \in \mathcal{F}_m, x \in \mathbb{R}^d, w \in P_M \right\}, \quad (5)$$

where $P_M = \{w \in \mathbb{R}^M : \sum_{m=1}^M w_m = 1, w_m \geq 0\}$ denotes the probability simplex. The gradient assistance weights $w_m$'s are interpreted as the contributions of each organization at a particular greedy update step. The gradient assistance weights are constrained to sum to one to ensure the function space is compact and the solutions exist.

We propose the following solution so that *each organization can operate on its own local data, model, and objective function*. Alice initializes with a startup model, denoted by $F^0(x) = F^0(x_1, y_1)$, based only on her local data and labels. Alice broadcasts $r_1$ (named 'pseudo residuals') to each organization

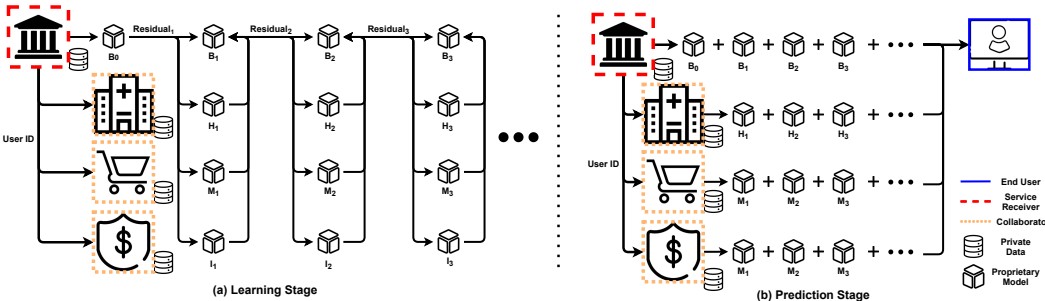

Figure 3: Learning and Prediction Stages for Gradient Assisted Learning (GAL).

$m$, $m = 2, \cdots, M$, who will then fit a local model $f_m$ using $r_1$. Each organization will then send the fitted values from $f_m$ to Alice, who will train suitable gradient assistance weights $w_m$. Subsequently, Alice finds the $\eta$ in (4) that minimizes her current empirical risk. The above procedure is iterated for a finite number of rounds until Alice obtains a satisfactory performance (e.g., on validation data). The validation will be based on the same technique as the prediction stage to be described below. This training stage is described under the 'learning stage' of Algorithm 1. Note that the pseudocode is from the perspective of Alice, the service receiver. Each organization $m$ will only need to perform the empirical risk minimization using the label $r_1^t$ sent by Alice at each round $t$.

In the Prediction/Inference stage (given above in Algorithm 1), other organizations send prediction results generated from their local models to Alice, who will calculate a prediction result $F^T(x)$ that is implicitly operated on $x$, where $T$ is the number of iteration steps.

The idea of approximating functional derivatives with regularized functions was historically used to develop the seminal work of gradient boosting [9, 10]. The above method reduces to the standard gradient boosting algorithm when there is only one organization.

Organizations in our learning framework form a shared community of interest. Each service-providing organization can provide end-to-end assistance for an organization without sharing anyone's proprietary data, models, and objective functions. In practice, the participating organizations may receive financial rewards from the one to assist. Moreover, every organization in this framework can provide its own task and seek help from others. As a result, all organizations become mutually beneficial to each other. We provide a realistic example in Figure 3 to demonstrate each step of Algorithm 1. We elaborate on the learning and prediction procedures in the Appendix.

We also provide an asymptotic convergence analysis for the GAL algorithm, where the goal is to minimize a loss $f \mapsto \mathcal{L}(f)$ over a function class through step-wise function aggregations. Because of the greedy nature of GAL, we consider the function class to be the linear span of organization-specific $\mathcal{F}_m$. The following result states that the GAL can produce a solution that attains the infimum of $\mathcal{L}(f)$. More technical details are included in the Appendix.

**Theorem 1** *Assume that the loss (functional) $f \mapsto \mathcal{L}(f)$ is convex and differentiable on $\mathcal{F}$, the function $u \mapsto \mathcal{L}(f + ug)$ has an upper-bounded second-order derivative $\partial^2 \mathcal{L}(f + ug)/\partial u^2$ for all $f \in span(\mathcal{F}_1, \ldots, \mathcal{F}_M)$ and $g \in \cup_{m=1}^{M} \mathcal{F}_m$, and the ranges of learning rates $\{a_t\}_{t=1,2,\ldots}$ satisfy $\sum_{t=1}^{\infty} a_t = \infty$, $\sum_{t=1}^{\infty} a_t^2 < \infty$. Then, the GAL algorithm satisfies $\mathcal{L}(F^t) \to \inf_{f \in span(\mathcal{F}_1, \ldots, \mathcal{F}_M)} \mathcal{L}(f)$ as $t \to \infty$, with a convergence rate at the order of $O(\sum_{\tau=1}^{t} (a_{1:\tau}/a_{1:t}) a_\tau^2)$.*

## 4   Experimental Studies

**Baselines** Our experiments are performed with four baselines, including 'Interm', 'Late', 'Joint', 'Alone', and 'AL'. 'Interm' and 'Late' refer to intermediate and late data fusions [7, 8], respectively. The intermediate data fusion ('Interm') sums up the intermediate features (output before the last layer) from each separate feature extractor, and then the aggregated feature is passed into a shared last layer to output the final prediction. The late data fusion ('Late') sums up the final prediction of each separate local model. Thus, 'Interm' works only for deep learning models such as CNN and LSTM by averaging the hidden representation of each local model, while 'Late' also works for Linear models as it aggregates the output of each local model. 'Joint' is the case where all the

data are held by Alice and trained with the Gradient Boosting reduced from GAL. 'Alone' is the single-agent scenario, where only Alice's data are used for learning and prediction. 'AL' represents the performance of Assisted Learning [23]. GAL is expected to perform close to the centralized baselines, including 'Interm', 'Late', and 'Joint' cases, while significantly outperforming the 'Alone' and 'AL' cases. The summary statistics of each dataset are elaborated in Table 7 of the Appendix. Details of learning hyper-parameters are included in Table 9 of the Appendix. We conducted four random experiments for all datasets with different seeds, and the standard errors are shown in the brackets of all tables.

## 4.1 Model Autonomy

Recall that GAL allows each organization to choose its own local model. We demonstrate the performance of autonomous local models with UCI datasets downloadable from the *scikit-learn* package [24], including Diabetes [25], Boston Housing [26], Blob [24], Iris [27], Wine [28], Breast Cancer [29], and QSAR [30] datasets, where we randomly partition the features into 2, 4, or 8 subsets. For all the UCI datasets, we train on 80% of the available data and test on the remaining.

Table 1: Results of the UCI datasets ($M = 8$) with Linear, GB, SVM and GB-SVM models. The Diabetes and Boston Housing (regression) are evaluated with Mean Absolute Deviation (MAD), and the rest (classification) are evaluated with Accuracy.

| Dataset | Model | Diabetes($\downarrow$) | BostonHousing($\downarrow$) | Blob($\uparrow$) | Wine($\uparrow$) | BreastCancer($\uparrow$) | QSAR($\uparrow$) |
|---|---|---|---|---|---|---|---|
| Late | Linear | 136.2(0.1) | 8.0(0.0) | 100.0(0.0) | 100.0(0.0) | 96.9(0.4) | 76.9(0.8) |
| Joint | Linear | 43.4(0.3) | 3.0(0.0) | 100.0(0.0) | 100.0(0.0) | 98.9(0.4) | 84.0(0.2) |
| Alone | Linear | 59.7(9.2) | 5.8(0.9) | 41.3(10.8) | 63.9(15.6) | 92.5(3.4) | 68.8(3.4) |
| AL | Linear | 51.5(4.6) | 4.7(0.6) | 97.5(2.5) | 95.1(3.6) | 97.7(1.1) | 70.6(5.2) |
| GAL | Linear | 42.7(0.6) | 3.2(0.2) | 100.0(0.0) | 96.5(3.0) | 98.5(0.7) | 82.5(0.8) |
| GAL | GB | 56.5(2.8) | 3.8(0.5) | 96.3(2.2) | 95.8(1.4) | 96.1(1.0) | 84.8(0.9) |
| GAL | SVM | 46.6(1.4) | 2.9(0.2) | 96.3(4.1) | 96.5(1.2) | 99.1(1.1) | 85.5(0.7) |
| GAL | GB-SVM | 49.8(2.6) | 3.4(0.8) | 70.0(7.9) | 95.8(1.4) | 93.2(1.6) | 82.9(1.5) |

We experiment with Linear, Gradient Boosting (GB), and Support Vector Machine (SVM) for local models $f_m(\cdot)$ with the UCI datasets. We also demonstrate the performance of the scenario (GB-SVM) where half of the organizations use GB and the other half uses SVM. The experimental results are shown in Tables 1. $\downarrow$ indicates the smaller the better, while $\uparrow$ indicates the larger the better. Our method significantly outperforms the baselines 'Alone' and 'AL.' The results also demonstrate that with GAL, an organization with little informative data and free choice of its local model (model autonomy) can leverage others' local data and models and even achieve near-oracle performance. We point out that although 'Interm,' 'Late,' and 'Joint' marginally outperform our method, they require training from centralized data. Our GAL algorithm replaces the true labels used in 'Interm,' 'Late,' and 'Joint' centralized cases with pseudo-residuals. The results from both regression and classification datasets with various model settings lead to similar conclusions.

## 4.2 Deep Model Sharing

We demonstrate that our method is effective for deep models by using MNIST [31] and CIFAR10 [32] image datasets, where we split each image into patches as depicted in Figure 6. We use Convolutional Neural Networks (CNN) for both datasets, and the model architecture can be found in Table 8 of the Appendix. We visualize the performance of CIFAR10 at each assistance round in Figure 4 (a-c). The number of assistance rounds needed to approach the centralized performance is small (e.g., often within ten). The experimental results are shown in Tables 2. GAL significantly outperforms the bottom line 'Alone' in all the settings. This is expected since the first organization holds partial data and does not receive any assistance under 'Alone.' Interestingly, the performance of MNIST for $M = 8$ drops significantly under 'Alone' because the organization only holds the left upper image patch, which is usually completely dark, as shown in Figure 6.

Table 2: Results of the MNIST and CIFAR10 ($M = 8$) datasets with CNN model. The MNIST and CIFAR10 are evaluated with Accuracy. GAL$_{DMS}$ represents the results with Deep Model Sharing. More results for $M = 2$ and $4$ are in the Appendix.

| Dataset | MNIST($\uparrow$) | CIFAR10($\uparrow$) |
|---|---|---|
| Interm | 98.8(0.1) | 78.2(0.2) |
| Late | 98.0(0.1) | 74.4(0.3) |
| Joint | 99.4(0.0) | 80.1(0.2) |
| Alone | 24.2(0.1) | 46.3(0.3) |
| AL | 34.3(0.1) | 51.1(0.2) |
| GAL | 96.3(0.6) | **74.3(0.2)** |
| GAL$_{DMS}$ | **96.3(0.5)** | 67.0(0.3) |

Because local deep learning models such as CNN can consume extensive computation space, we propose Deep Model Sharing (DMS) to allow sharing feature extractors of deep models across all iterations to save memory. In particular, we propose to jointly train the feature extractors with residuals from previous iterations as well as from the current iteration by adding an additional last prediction layer. The local deep model $f_m^t(\cdot)$ is composed of $f_{m,o}^t(f_{m,e}(\cdot)))$, where $f_{m,o}^t(\cdot)$ is the last output layer at assistance round $t$ and $f_{m,e}(\cdot)$ is the deep feature extractor shared across multiple assistance rounds. For each assistance round, local organizations fits pseudo-residuals across previous assistance rounds $r_1^{1:t}$ with $f_m^{1:t} = \mathrm{argmin}_{f_m \in \mathcal{F}_m} \mathbb{E}_N \ell_m \left( r_1^{1:t}, f_{m,o}^{1:t}(f_{m,e}(x_m)) \right)$. It is worth mentioning that we do not expect such trade-off $\mathrm{GAL_{DMS}}$ to consistently outperform AL and GAL because a single feature extractor may not well fit residuals across many iterations. The results in Table 2 show that sharing the feature extractor across multiple assistance rounds can still outperform the 'Alone' case. Thus, DMS can provide a trade-off between predictive performance and computation space. The detailed comparisons can be found in Table 14.

## 4.3 Comparison with AL

As demonstrated in our extensive experiments, the proposed GAL outperforms AL in terms of predictive performance. In particular, AL converges not only worse but also slower than GAL. This is due to the fact that AL uses a constant assisted learning rate and trains participating organizations in a sequential manner. Moreover, sequentially training participating organizations also requires much more computation and communication overhead. We compare the computation and communication complexity between AL and GAL under the constraint of the same communication cost as demonstrated in Table 14. Because AL sequentially trains each organization while GAL allows organizations to train locally in parallel, the computation time and communication round of AL is $M\times$ those of GAL. The GAL with Deep Model Sharing ($\mathrm{GAL_{DMS}}$) saves $T\times$ computation space by sharing the feature extractor of deep models. In summary, GAL generalizes the problem scope, reduces the computation and communication complexity, and achieves significantly better results.

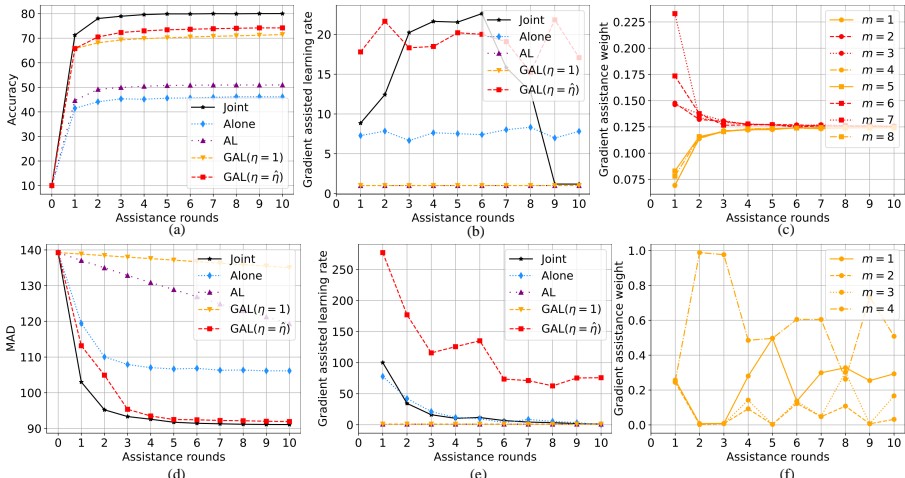

Figure 4: Results of the CIFAR10 (a-c) ($M = 8$) and MIMICL (d-f) ($M = 4$) datasets. GAL significantly outperforms 'Alone' and 'AL'. Our method also performs close to the centralized baselines. The gradient assisted learning rate diminishes to zero as the overarching loss converges. A constant gradient assisted learning rate ($\eta = 1$) converges much slower. The gradient assistance weights exhibits interpretability of the importance of organizations as the weights of the central image patches ($m = \{2, 3, 6, 7\}$) of CIFAR10 dataset are larger than the boundary patches ($m = \{1, 4, 5, 8\}$) in the first few rounds. More results can be found in the Appendix.

## 4.4 Case Studies

The results in Table 3 demonstrate the utility of GAL in various practical applications. We illustrate the results across multiple assistance rounds of MIMICL in Figure 4 (d-f). GAL significantly outperforms 'Alone' and 'AL.' Our method also performs close to the centralized baselines.

Table 3: Results of case studies of 3D object recognition and medical time series forecasting.

| Dataset | ModelNet40($\uparrow$) | ShapeNet55($\uparrow$) | MIMICL($\downarrow$) | MIMICM($\uparrow$) |
|---|---|---|---|---|
| Interm | 75.3(18.2) | 88.6(0.1) | 64.6(0.9) | 0.90(0.0) |
| Late | 86.6(0.2) | 88.4(0.1) | 71.4(0.2) | 0.91(0.0) |
| Joint | 46.3(1.4) | 16.3(0.0) | 91.1(0.7) | 0.82(0.0) |
| Alone | 76.4(1.1) | 81.3(0.6) | 106.1(0.3) | 0.78(0.0) |
| AL | 77.3(2.8) | 83.8(0.0) | 119.3(0.3) | 0.86(0.0) |
| GAL | 83.0(0.2) | 84.1(0.6) | **91.9(2.3)** | **0.88(0.0)** |
| GAL$_{\text{DMS}}$ | **83.2(0.3)** | **85.3(0.2)** | 97.7(2.9) | 0.81(0.0) |

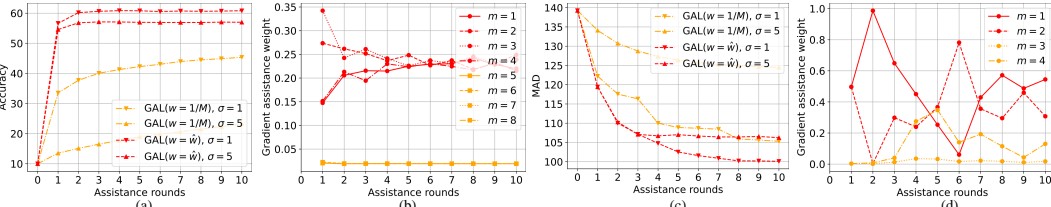

Figure 5: Ablation study results on CIFAR10 (a-b) ($M = 8$) and MIMICL (c-d) ($M = 4$) datasets. Plots (a,c) show that the GAL equipped with gradient assistance weight significantly outperforms the GAL with direct average under noise injections ($\mathcal{N}(0, \sigma^2)$, $\sigma = \{1, 5\}$) to the transmitted pseudo-residual to half of the organizations during learning and prediction. Plots (b,d) show the gradient assistance weight of noisy (in orange, $\sigma = 1$) and noise-free organizations (in red).

**Three-dimensional object recognition** We use the shape representation of 3D objects for recognition from a collection of rendered views on 2D images. We generate $M = 12$ 2D camera views of ModelNet40 [33] and ShapeNet55 [34] datasets following [35]. Cameras are treated as decentralized learners in our experiments. We adopt the same CNN architecture used for MNIST and CIFAR10 datasets. It is worth mentioning that 'Joint' of ModelNet40 and ShapeNet55 performs considerably worse than other baselines because 'Joint' uses a single CNN feature extractor to process all images from twelve angles [35].

**Medical time series forecasting** We use the in-hospital dataset MIMIC3 [36], where the task aims to predict the Length-of-stay (MIMICL) and Mortality rate (MIMICM) of the ICU stays of patients. We follow the benchmark work [37] to process raw data and split the features into four organizations, including 1) microbiology measurement, 2) demographic information, 3) body measurement, and 4) International Classification of Diseases (ICD). We use MAD to evaluate the result of MIMICL (regression) and the Area Under the Curve-Receiver Operating Characteristics (AUROC) to evaluate the result of MIMICM (imbalanced binary classification). Our backbone model (LSTM) is the same as the one used in the benchmark work [38]. It is worth noting that the data features among the four organizations are aligned with time stamps which is a natural identifier for time series forecasting.

## 4.5 Ablation Studies

**Local objective functions** Our method does not require local regression loss functions $\ell_m$ to be shared with other organizations, but they are supposed to be beneficial for assisting Alice. We conduct ablation studies on the choice of various local regression loss functions. In this study shown in Table 4, we use different regression functions $\ell_q(y, \hat{y}) = |y - \hat{y}|^q$ where $q \in \{1, 1.5, 2, 4\}$. $(\ell_{q_1}, \ell_{q_2})$ allows half of the organizations to use $\ell_{q_1}$ while the other half to use $l_{q_2}$. The results show that the classification task generally works better with $q > 1$.

Table 4: Ablation study on the local objective function ($M = 8$). More results are in the Appendix.

| Dataset | Diabetes($\downarrow$) | BostonHousing($\downarrow$) | Blob($\uparrow$) | Wine($\uparrow$) | BreastCancer($\uparrow$) | QSAR($\uparrow$) | MNIST($\uparrow$) | CIFAR10($\uparrow$) |
|---|---|---|---|---|---|---|---|---|
| Alone | 59.7(9.2) | 5.8(0.9) | 41.3(0.0) | 63.9(0.0) | 92.5(0.4) | 68.8(0.2) | 24.2(0.1) | 46.3(0.3) |
| $\ell_1$ | **42.7(0.6)** | 3.2(0.2) | 42.5(12.5) | 95.1(4.1) | 97.4(1.1) | 64.3(3.6) | 90.5(1.9) | 27.8(1.7) |
| $\ell_{1.5}$ | 43.4(1.0) | **2.9(0.1)** | 100.0(0.0) | 95.8(4.2) | 97.4(0.6) | 80.2(1.3) | 94.5(0.1) | 70.2(0.8) |
| $\ell_2$ | 44.8(1.9) | 3.0(0.1) | **100.0(0.0)** | **96.5(3.0)** | 98.5(0.7) | **82.5(0.8)** | 96.3(0.6) | **74.3(0.2)** |
| $\ell_4$ | 45.8(1.3) | 3.2(0.2) | 97.5(4.3) | 98.6(1.4) | **99.1(0.9)** | 81.3(0.7) | **98.1(0.1)** | 73.2(0.3) |
| $(\ell_1, \ell_2)$ | 43.3(1.5) | 3.2(0.3) | 100.0(0.0) | 96.5(3.0) | 96.7(0.7) | 80.0(1.0) | 94.6(1.1) | 65.0(0.2) |

**Privacy enhancement** Our learning framework does not require organizations to share local data, models, and objective functions. One potential limitation of our approach is that assisting organizations may infer Alice's information based on the shared pseudo-residuals. Therefore, we suggest to further enhancing privacy by adopting the framework of Differential Privacy (DP) [39] or Interval Privacy (IP) [40]. We use the Laplace mechanism with $\alpha = 1$ for DP and set the number of intervals of IP to be 1. We add a moderate amount of noise to the pseudo-residuals in hindsight. In Tables 5, we demonstrate that privacy-enhanced GAL can still outperform the 'Alone' case.

Table 5: Ablation study on the privacy enhancement (maximal $M$). GAL$_{DP}$ and GAL$_{IP}$ represent privacy-enhanced by DP and IP, respectively.

| Dataset | Diabetes(↓) | BostonHousing(↓) | Blob(↑) | Wine(↑) | BreastCancer(↑) | QSAR(↑) | MNIST(↑) | CIFAR10(↑) | ModelNet40(↑) | ShapeNet55(↑) | MIMICL(↓) | MIMICM(↑) |
|---|---|---|---|---|---|---|---|---|---|---|---|---|
| Alone | 59.7(9.2) | 5.8(0.9) | 41.3(10.8) | 63.9(15.6) | 92.5(3.4) | 68.8(3.4) | 24.2(0.1) | 46.3(0.3) | 76.4(1.1) | 81.3(0.6) | 106.1(0.3) | 0.78(0.0) |
| GAL$_{DP}$ | 52.2(1.0) | 4.3(1.1) | 51.3(10.8) | 88.2(7.9) | 94.7(1.1) | 80.5(1.8) | 94.3(0.6) | 56.8(0.7) | 46.6(2.8) | 46.7(7.5) | 94.9(3.2) | 0.59(0.0) |
| GAL$_{IP}$ | 51.8(0.7) | 4.2(1.1) | 100.0(0.0) | 95.8(3.1) | 96.1(0.8) | 84.8(0.9) | 94.7(0.5) | 69.2(0.1) | 59.8(0.7) | 58.0(2.5) | 95.5(4.9) | 0.59(0.0) |

**Gradient assisted learning rate** We illustrate the gradient assisted learning rate of CIFAR10 and MIMICL datasets at each assistance round in Figure 4(b,e). We perform a line search for the gradient assisted learning rate with the Limited-Memory BFGS optimizer, which improves the convergence rates compared with SGD and Adam. We conduct an ablation study using a constant gradient assisted learning rate ($\eta = 1$). As shown in Figure 4(a,d), the constant gradient assisted learning rate leads to a convergence much slower than the line-search method. Fast convergence is desirable since the computation and communication cost increases with the number of assistance rounds. To determine the maximal number of assistance rounds $T$ for the service receiver, we can run the GAL procedure until the gradient assisted learning rate becomes small. When the gradient assistance rate is low, as shown in Figure 4(e), the overarching loss converges to zero. In this light, an organization may stop receiving assisted learning when the gradient assisted learning rate is below a threshold.

**Gradient assistance weights** We show the gradient assistance weights of CIFAR10 and MIMICL datasets at each assistance round in Figure 4(c,f). The results of MNIST and CIFAR10 show that the gradient assistance weights exhibit interpretability of the importance of organizations because the image patches with dominant contributions are $m = (2, 3, 6, 7)$ (colored in red). These image patches correspond to the center of the original image, which matches our intuition appealingly. The weights converge to uniform as the residuals of later iterations are small. We also conduct an ablation study of the gradient assistance weights in Figure 5 by adding noises (Gaussian with zero mean and $\sigma^2$ variance, $\sigma \in \{1, 5\}$) to the output of a randomly chosen half of the clients during learning and prediction. Adding noise simulates realistic scenarios where some assisting organizations may be noninformative or inject noise. We summarize the ablation study results in Table 6. The results show that the GAL with gradient assistance weights is more robust than the GAL with a direct average.

Table 6: Ablation study (maximal $M$) of gradient assistance weights by adding noises to the predicted outputs from half of the organizations.

| Noise | Weight | Diabetes(↓) | BostonHousing(↓) | Blob(↑) | Wine(↑) | BreastCancer(↑) | QSAR(↑) | MNIST(↑) | CIFAR10(↑) | ModelNet40(↑) | ShapeNet55(↑) | MIMICL(↓) | MIMICM(↑) |
|---|---|---|---|---|---|---|---|---|---|---|---|---|---|
| $\sigma = 1$ | ✗ | 49.0(1.6) | 4.3(0.2) | 46.3(6.5) | 81.2(5.3) | 90.8(2.5) | 73.2(1.0) | 75.1(0.4) | 45.4(0.3) | 55.8(1.0) | 67.6(0.8) | 105.3(0.6) | 0.54(0.0) |
| | ✓ | **46.4(2.3)** | **4.0(0.2)** | **78.8(8.2)** | **88.9(2.0)** | **96.7(1.0)** | **78.9(1.2)** | **92.7(0.1)** | **61.0(0.4)** | **78.3(0.4)** | **80.7(0.3)** | **100.1(0.6)** | **0.75(0.0)** |
| $\sigma = 5$ | ✗ | 61.0(2.4) | 5.8(0.2) | 12.5(2.5) | 54.2(6.9) | 78.5(2.0) | 61.8(0.5) | 33.8(0.3) | 23.3(0.6) | 24.5(0.9) | 39.6(0.9) | 124.5(0.0) | 0.52(0.0) |
| | ✓ | **49.7(3.1)** | **4.7(0.5)** | **62.5(9.0)** | **84.7(1.4)** | **96.9(1.3)** | **77.1(0.8)** | **92.1(0.2)** | **57.3(0.3)** | **77.5(0.4)** | **79.8(0.4)** | **108.3(3.5)** | **0.67(0.0)** |

## 5 Conclusion

We proposed Gradient Assisted Learning, a decentralized learning method for multiple organizations to collaborate without sharing data, models, and objective functions. The proposed method can significantly outperform the local learning baselines and achieve near-oracle performance as if data were centralized on various datasets. All participants form a shared community of interest by autonomously building their own model and iteratively fitting the gradients of the overarching objective function. We also demonstrate asymptotic convergence analysis and practical case studies of GAL. Moreover, this is achieved without any constraints on the models selected by the collaborating organizations.

## Acknowledgments

The work of Enmao Diao and Vahid Tarokh was supported by the Office of Naval Research (ONR) under grant number N00014-18-1-2244. The work of Jie Ding was supported by the National Science Foundation (NSF) under grant number ECCS-2038603.

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
