# OpenReview forum: "GAL: Gradient Assisted Learning for Decentralized Multi-Organization Collaborations"
_NeurIPS.cc/2022/Conference — NeurIPS 2022 Accept_

### Official Review · Reviewer_HvzM · 2022-06-23

**Rating:** 6
**Confidence:** 4
**Soundness:** 3 good
**Presentation:** 3 good
**Contribution:** 3 good

**Summary:**

Gradient Assisted Learning (GAL), a decentralized collaborative supervised learning methodology, is proposed. GAL works to assist a particular organization (dubbed "Alice") with their learning problem by utilizing other organization's learning capabilities but without sharing any sensitive information such as models or raw data between any two organizations. This is accomplished by iteratively having each "helper" organization learn a model of the (pseudo-)residual using their own data and sending the model's prediction back to Alice. At each iteration, Alice updates the overall model to be learned by incorporating these predictions and optimizing their associated weightings. At test-time, Alice queries the helper organizations for their predictions and feeds those predictions through the learned weightings to arrive at a prediction. A theorem is given that states that, under mild technical conditions, the loss of the GAL model converges (as the number of "assistance iterations" approaches infinity) to the optimal loss amongst all models in the span of the organizations' model function classes. Experiments on a variety of datasets shows that GAL outperforms prior state-of-the-art decentralized methods, with performance approaching that of centralized learning paradigms.

**Questions:**

Concerns/Suggestions:

CS1. Please explicitly tell the reader what the arrows in the tables represent.

CS2. The authors mention the "Interm" method in relation to Tables 1, 8, and 9, but this method does not appear in these tables.

CS3. In the proof of the theorem, when you multiplied both sides of (12) by $|\mu_{f^*}^g-\mu_{F^{t-1}}^g|$ and then summed up to arrive at (13), it appears to the reviewer that you forgot to multiply the $a_t^2$ term---shouldn't (13) have a factor of $\lVert\mu_{f^*}-\mu_{F^{t-1}}\rVert_1$ attached to the $a_t^2$ term? This may cause issues with the convergence of this term to zero as $t \to \infty$; you may need a condition requiring $a_t^2$ to go to zero faster than the partial sums of $a_t$ goes to $\infty$.

CS4. The authors state that "Prior work on AL is limited to mean squared loss regression with only two organizations using a sequential exchange of information." This doesn't appear to be entirely true---the original AL methodology appears applicable to regression tasks with $m\in\mathbb{N}$ organizations using arbitrary models and loss functions (not just mean squared loss), so long as the loss functions are minimized at the origin (e.g., they are norms). What is true is that the theoretical guarantee for converging to the oracle performance given in the AL paper is restricted to all $m$ organizations using linear regression models. It so happens that the pseudo-residual GAL uses, i.e., the functional gradient step, reduces to the conventional residual in the regression setting when Alice's loss function is equal to the squared error loss. With respect to this limitation, GAL generalizes past the regression task setting (allowing for classification settings) by sharing these pseudo-residuals (gradient values) instead of the residuals used in AL. Please fix these overstatements.

CS5. The reviewer is concerned with some of the claims made in Section 4.3 "Comparison with AL." In particular, the authors argue that GAL's enhanced performance over AL is in part due to the parallelizability (with respect to the organizations) of GAL, whereas AL is trained sequentially. However, AL appears to be immediately parallelizable as well; the `for` loop over organizations $j$ in Procedure 1 of the AL paper [1] does not depend on organization $j' \ne j$, so one may simply perform the AL model fitting for organization $j$ in parallel from the model fitting for the other organizations. With this in mind, GAL is not necessarily $M\times$ faster than AL as claimed in the paper, so the impact of the authors' Table 12 is limited to the contributions of "deep model sharing," which the authors claim is not always expected to help the performance of GAL.

CS6. Overall, the reviewer found the contributions to be overstated with respect to the earlier paper [1].


Questions:

Q1. Can you give an actual rate of convergence in your theorem instead of just an asymptotic result?


[1] Xian, Xun, et al. "Assisted learning: A framework for multi-organization learning." Neurips 2020.

**Limitations:**

The authors do a good job at describing their limitations, namely, that their theoretical convergence result is not sufficiently strong to describe why GAL converges after only a few iterations of "assistance," and that it may be possible to infer sensitive information from the shared pseudo-residuals. The authors demonstrate that upon adding moderate amounts of noise to the pseudo-residuals to take care of the latter limitation, the performance of GAL does not significantly deteriorate, and that performance is still better than that without collaboration. However, it would be interesting to discuss (and potentially test) the issue of adversarial sensitivity; what happens if one of the organizations is malicious or subject to an adversarial attack, either in their data or in the predictions that they feed back to Alice? Relative to traditional adversarial machine learning settings (wherein the adversary is typically assumed access to just one mode of attack at a time, e.g., the input data at test time), the proposed decentralized learning paradigm potentially offers an adversary more avenues for attack, e.g., at a particular organization's input data and at the various predictions fed to Alice.

**Strengths And Weaknesses:**

Strengths:

S1. Quality. The authors formulate their methodology into a technically sound and rigorous mathematical and algorithmic framework. The experiments are thorough and demonstrate the strengths of the proposed method in a wide range of applications.

S2. Clarity. The paper is well-motivated and easy to read.


Weaknesses:

W1. Originality. The proposed methodology is extremely similar to the prior "assisted learning" (AL) work [1], providing what seems to be only incremental extensions to the technical aspects of the AL framework. See below for more details regarding this concern.


Neutral:

N1. Significance. Since the proposed method (GAL) applies to more general problems (e.g., classification) than AL and is shown to outperform AL in the considered benchmarks, the reviewer finds that the paper advances the practical strength and performance of the literature's decentralized collaborative learning methods. However, the reviewer believes that the approaches taken by the authors to extend AL to GAL (i.e., replacing regression residuals with classification/regression pseudo-residuals, and optimizing the model weights and learning rates) are not significant conceptual/theoretical advancements.


[1] Xian, Xun, et al. "Assisted learning: A framework for multi-organization learning." Neurips 2020.

---

> ### Author Response · Authors · 2022-08-02
> **Response**
>
> Thank you for your time and constructive comments. We have addressed all the comments below. The following major changes will be included in the revision. We hope the responses and planned revisions will be viewed favorably.
>
> 1. Originality and Significance
> > To address your concern and the following comments, we rephrased the discussions regarding AL and GAL accordingly. We also highlighted the following contributions of GAL that we believe are significant. 1) AL was developed from a linear projection perspective, while GAL was from a gradient boosting perspective. 2) Inspired by gradient boosting, GAL introduced pseudo-residuals to generalize the overarching loss at the functional gradient step to be any differentiable loss, while AL is limited to regression loss. 3) GAL introduced gradient assisted learning rate to obtain faster convergence and superior performance than AL. 4) GAL introduced gradient assistance weights to perform parallel aggregation of model predictions to reduce the run time. We also demonstrated that gradient assistance weights could enhance robustness against noisy participants. 5) GAL introduced deep model sharing to save the computation space for deep models. Overall, AL and GAL share similar motivations and concepts, but they have significant differences from methodological and theoretical perspectives.
>
> 2. Please explicitly tell the reader what the arrows in the tables represent.
> > Thanks for pointing out this. We have clarified these arrows in the tables in Section 4.1 (highlighted in the revision).
>
> 3. The authors mention the "Interm" method in relation to Tables 1, 8, and 9, but this method does not appear in these tables.
> > The ``Interm'' method was not used for the UCI datasets due to its nature. We used linear models for UCI datasets. But the `Interm' method requires each partition to use a backprobable model and aggregate the features before the last layer. We describe this discrepancy in Section 4 Baselines (highlighted in the revision).
>
> 4. In the proof of the theorem, when you multiplied both sides of (12)...
> > Thank you for pointing out this error. Yes, we forgot to multiply $\mu_{f^*}^g - \mu_{F^{t-1}}^g$ to the term $a_t^2$ in (13). Also, we have checked that the follow-up inequality (14) does not need to be modified, and there is no need for an additional condition to guarantee convergence.
>
> 5. The authors state that "Prior work on AL is limited to ...
> > Thank you for pointing this out. We revised the statement about the AL in Section 2 (highlighted in the revision) as ``The original AL methodology applies to regression tasks. It is derived from a linear projection perspective, and its convergence to the oracle performance was theoretically justified for linear regression models with quadratic loss. Inspired by Gradient Boosting, the proposed Gradient Assisted Learning (GAL) is a general method for multiple organizations to assist each other in supervised learning scenarios. Overall, AL and GAL share similar motivations and concepts but significantly differ from methodological and theoretical perspectives. More specifically, ...''  In our experiments, we also used pseudo-residuals in classification tasks for AL to have a fair comparison. We found that AL performs worse than GAL because it sequentially fits the pseudo-residuals and does not line search for the gradient assisted learning rate. The ablation studies shown in Figure 4 (a,d) also corroborated the effectiveness of parallel aggregation and optimizing gradient assisted learning rate.
>
> 6. The reviewer is concerned with some of the claims made in Section 4.3 "Comparison with AL."...
> > We think GAL is parallelizable because the introduced gradient assistance weights explicitly aggregate the predicted outputs from each organization in each round, which naturally came from a gradient boosting perspective in Equations (4)(5) of our paper. Although Procedure 1 of AL may run in parallel, it is not clear how we can combine the outputs of organizations. The formulation of linear projection in AL does not directly provide such an aggregation mechanism.
>
> 7. Overall, the reviewer found the contributions to be overstated with respect to the earlier paper [1].
> > Thank you for pointing out this concern. Following this and earlier comments, we have rewritten the related statements to reduce the overstatement and highlight the uniqueness of GAL. Please let us know if the current version reads well.
>
> 8. Can you give an actual rate of convergence in your theorem instead of just an asymptotic result?
> > Thank you for this suggestion. Yes, we can obtain an explicit convergence rate. By revisiting the proof of Theorem~1, we found that the rate of convergence is at the order of $O(\sum_{\tau=1}^t (a_{1:\tau}/a_{1:t})  a_{\tau}^2)$. We have revised the theorem statement in Appendix B (highlighted in the revision).

---

> > ### Author Response · Authors · 2022-08-02
> > **Response continue**
> >
> > 9. However, it would be interesting to discuss (and potentially test) the issue of adversarial sensitivity...
> > > Thank you for the suggestions. We address your excellent point of adversarial sensitivity in the following two ways. First, we experimentally studied the sensitivity of learning results to contaminated predicted outputs in Appendix D.4.2 (highlighted in the revision). Unlike adding noise to the transmitted pseudo-residuals to enhance privacy in Appendix D.4.1, the experimental studies in Appendix D.4.2 demonstrate that the gradient assistance weights can effectively promote robustness against noisy organizations. Nevertheless, we understand that this is not an adversarial setting where perturbations are not random noise but crafted in a way adaptive to particular models or learning tasks. Second, following your comment, we have discussed several interesting adversarial problems related to our work in Appendix A.2 of the revision (highlighted). Specifically, we have discussed possible research problems communed to adversarial examples, poisoning attacks, backdoor attacks, and model-stealing attacks.

---

> > > ### Comment · Reviewer_HvzM · 2022-08-05
> > > **Updated Review**
> > >
> > > The authors have addressed my primary concerns. Namely, the Theorem's convergence rate, the revision of contribution overstatements, the expanded comparison between the proposed method and the prior AL method, and the discussion on relation to adversarial settings and potentials for future work in this realm are all nice revisions to the paper. I have increased my score by one point. There remain two small issues I hope to see implemented in the second revision:
> > >
> > > 1. In line 227 of the revised manuscript, there seems to be a typo; it should state "...while $\uparrow$ indicates..." instead of "...while $\downarrow$ indicates...".
> > >
> > > 2. The notation $1:t$ (specifically, $a_{1:t}$ in Theorem 1, and $f_m^{1:t}$ and $r_1^{1:t}$ in Section 4.2) does not appear to be defined in the main body of the paper (although I do see that it is defined for $a_{1:t}$ in the proof of Theorem 1). Please clearly define what these notations mean in the Notations section, Section 3.1, before they are used in the main body of the paper.

---

> > > > ### Author Response · Authors · 2022-08-06
> > > > **Thank you**
> > > >
> > > > Thanks once again for your constructive comments. We have addressed the issues mentioned in your previous comment (highlighted in our second revision).

---

### Official Review · Reviewer_mrR4 · 2022-07-11

**Rating:** 6
**Confidence:** 3
**Soundness:** 3 good
**Presentation:** 3 good
**Contribution:** 3 good

**Summary:**

This paper proposes the gradient assistive learning (GAL) algorithm for a multi-organization collaborative learning setting. The proposed algorithm can be considered as a distributed version of the gradient boosting algorithm. The authors provide convergence analysis and experimental results to show the benefit of the proposed algorithm.

**Questions:**

What would be a more realistic setting to apply this method in the image / language modality?

**Limitations:**

Authors mentioned that they "do not foresee any negative societal impacts". I agree with this.

**Strengths And Weaknesses:**

Strengths:

The paper studies an interesting distributed learning scenario which I think is relatively less explored in the community. The proposed algorithm is a reasonable extension of prior work on assistive learning. This paper also builds a connection between the GAL algorithm and the classic gradient boosting algorithm. The paper is relatively well-written and easy to follow. Authors provide results on a variety of datasets. I also appreciate that the authors provided detailed descriptions of the experiments in the appendix, as well as the results in the privacy enhancement setting. I did not check the proof details but the theoretical result looked sound.
Overall, I think this is a good paper.

Weaknesses:

For the MNIST and CIFAR experiments, authors assume that the images are split into patches, and different patches are stored in different organizations. I don't think this is a realistic experimental setup, and it's unclear why different organizations only store a patch of a natural image. I would like to see a more realistic scenario where this method can be applied in deep learning models.

---

> ### Author Response · Authors · 2022-08-02
> **Response**
>
> Thank you for your time and constructive comments. We have addressed all the comments below. The following major changes will be included in the revision. We hope the responses and planned revisions will be viewed favorably.
>
> 1. For the MNIST and CIFAR experiments, authors assume that the images are split into patches, and different patches are stored in different organizations. I don't think this is a realistic experimental setup, and it's unclear why different organizations only store a patch of a natural image. I would like to see a more realistic scenario where this method can be applied in deep learning models.
> > We agree that the MNIST and CIFAR experiments are artificially created to prove concepts. We provided more realistic applications in Section 4.4 ``Case Studies''. In particular, we considered the applications of three-dimensional object recognition and medical time series forecasting. In the three-dimensional object recognition task, we have $M=12$ two-dimensional camera views of three-dimensional objects from ModelNet4 and ShapeNet55 datasets for classification. In the medical time series forecasting problem, we have time series features naturally collected from $M=4$ different departments: 1) microbiology measurements, 2) demographics, 3) body measurements, and 4) International Classification of Diseases (ICD), to predict the Length-of-stay (MIMICL) and Mortality rate (MIMICM) in ICU. We hope these experiments can provide more realistic demonstrations of the proposed approach.
>
> 2. What would be a more realistic setting to apply this method in the image / language modality?
> > Training with 2D image views captured from 3D objects is a realistic example of image modality. One may use it for decentralized object detection in application domains such as drone-based geological surveying, sensor monitoring, and autonomous mobility. Our method can also be used for time series data as the decentralized time series features can be naturally aligned with time stamps (e.g., for collaborative supply chain forecasting).

---

> ### Author Response · Authors · 2022-08-06
> **A kind reminder**
>
> Dear Reviewer mrR4,
>
> We would like to thank you again for the time you dedicated to reviewing our paper and your valuable comments. We believe that we have addressed your concerns. Since the end of discussion period is getting close and we have not heard back from you yet, we would appreciate if you kindly let us know of any other concerns you may have, and if we can be of any further assistance in clarifying any other issues.
>
> Thanks a lot again, and with sincerest best wishes
>
> Authors

---

> ### Author Response · Authors · 2022-08-09
> **A kind reminder**
>
> Dear Reviewer mrR4,
>
> We apologize for any inconvenience that our message may cause in advance. Again, we would like to thank you for the time you dedicated to reviewing our paper and your valuable comments. We believe that we have addressed your concerns. Since the end of discussion period is close and we have not heard back from you yet, we would appreciate if you kindly let us know of any concerns you may have, and if we can be of any further assistance in clarifying any other issues. We humbly remain at your disposal.
>
> Thanks a lot again, and with best wishes,
>
> Authors

---

### Official Review · Reviewer_XaCz · 2022-07-11

**Rating:** 5
**Confidence:** 4
**Soundness:** 2 fair
**Presentation:** 3 good
**Contribution:** 2 fair

**Summary:**

This paper studies collaborative learning between organizations without sharing data, models, and even objective functions. The authors propose a Gradient Assisted Learning (GAL) method that collaboratively optimize the aggregation of local loss functions, while each worker iteratively fits the gradients of the overarching objective function. The authors prove that the proposed GAL can achieve the performance close to centralized learning when all data, models, and objective functions are fully disclosed. The empirical results support this argument.

**Questions:**

Please response the cons in the rebuttal.

**Limitations:**

Limitations are duely discussed, while no potential negative societal impact is identified.

**Strengths And Weaknesses:**

Pros:
+ This paper addresses a real and timely problem that collaboratively optimize the aggregation of local loss functions of different organizations, while each worker iteratively fits the gradients of the overarching objective functions.
+ The paper is well-written and easy to follow.

Cons:
- I find the setting is a bit unrealistic/confusing and needs further clarification. Is there any restrictions on the discrepancy between the data distributions of different organizations? In some extreme cases, would it be possible that the gradients from other devices cannot assist or have bad effects?
- The experiments are not comprehensive enough. Does any other existing method can be applied here. I understand that these methods are not designed to this setting and they may be rigorously reasonable, but I find this would be helpful to discuss them.

---

> ### Author Response · Authors · 2022-08-02
> **Response**
>
> Thank you for your time and constructive comments. We have addressed all the comments below. The following major changes will be included in the revision. We hope the responses and planned revisions will be viewed favorably.
>
> 1. I find the setting is a bit unrealistic/confusing and needs further clarification. Is there any restrictions on the discrepancy between the data distributions of different organizations? In some extreme cases, would it be possible that the gradients from other devices cannot assist or have bad effects?
> > Our setting does not restrict the data features or distributions of different organizations--they may hold distinct or partially overlapping variable sets. The potential learning gain is attained through sharing gradient-based information. We agree with you that it is likely that certain features of an organization or specific organizations as a whole do not have predictive power toward the target. We introduced the gradient assistance weights to identify the helpful organizations in each round adaptively. Our experimental studies demonstrate that the gradient assistance weights effectively promote robustness against poor-performing organizations. For example, in Appendix D.4.2 (highlighted in the revision), we added a small amount of noise to the predicted outputs from half of the organizations. We also showed that optimizing the gradient assistance weights can reduce the weights of those less informative organizations. For example, the result shown in Figure 4(c) demonstrates that the central image patches of CIFAR10 images are more informative than the borderline image patches.
>
> 2. The experiments are not comprehensive enough. Does any other existing method can be applied here. I understand that these methods are not designed to this setting and they may be rigorously reasonable, but I find this would be helpful to discuss them.
> > We added a discussion of Vertical Federated Learning (VFL) for vertically partitioned data in Section 2 (highlighted in the revision). VFL is a federated learning extension of the intermediate data fusion method. Specifically, VFL reduces to the `Interm' case in our experiments if there is no communication constraint. VFL requires locally backpropable models and very frequent batchwise synchronization of gradients at an intermediate layer because the local model at each client constitutes a part of the globally backpropable model. On the contrary, GAL allows model-agnostic local training, so organizations can freely use different machine learning models locally (e.g., neural networks, decision trees, SVMs). Furthermore, our experimental results show that GAL needs only a few communication rounds to converge (often within ten). This is particularly suitable for organizational learners since they may not want frequent synchronizations.

---

> > ### Comment · Reviewer_XaCz · 2022-08-07
> > **A further question**
> >
> > Thanks for your response. I am interested in a further question. As you have commented: "We agree with you that it is likely that certain features of an organization or specific organizations as a whole do not have predictive power toward the target." An additional experiment about the performance of your algorithm in this extreme (?) case would make this paper better.

---

> > > ### Author Response · Authors · 2022-08-08
> > > **Response**
> > >
> > > Thanks for your constructive comments. In Table 19-21 (highlighted in the revision), we conduct additional experiments for an extreme case where half of organizations have no predictive power for the target, i.e. data features sampled from $\mathcal{N}(0,1)$. The results show that gradient assistance weights can weight higher for the organizations having predictive power and thus improve the performance. We only add the results of the UCI datasets due to the time constraint. We will add the results of other datasets once those experiments are finished.

---

> ### Author Response · Authors · 2022-08-06
> **A kind reminder**
>
> Dear Reviewer XaCz,
>
> We would like to thank you again for the time you dedicated to reviewing our paper and your valuable comments. We believe that we have addressed your concerns. Since  the end of discussion period is getting close and we have not heard back from you yet, we would appreciate if you kindly let us know of any other concerns you may have, and if we can be of any further assistance in clarifying any other issues.
>
> Thanks a lot again, and with sincerest best wishes
>
> Authors

---

### Official Review · Reviewer_3PDZ · 2022-07-12

**Rating:** 7
**Confidence:** 3
**Soundness:** 2 fair
**Presentation:** 3 good
**Contribution:** 3 good

**Summary:**

This paper proposes an algorithm for multi-agent learning. By sharing gradient information but not data among agents, this algorithm can achieve collaborative learning while keeping data privacy. This paper also shows this algorithm can converge in limit, and provides supporting numerical experiments.

**Questions:**

N/A

**Strengths And Weaknesses:**

Strengths:

1. This algorithm solve the feature-distributed optimization problem, which is valuable and practical.

2. The algorithm shares gradient information rather than data, which preserves the privacy.

3. This paper proves the algorithm converges in limit.

4. The numerical experiments are comprehensive and convincing.

Weaknesses:

1. The theoretical analysis may be able to be improved. The current proof only shows convergence in limit, maybe can obtain an explicit convergence rate.

---

> ### Author Response · Authors · 2022-08-02
> **Response**
>
> Thank you for your time and constructive comments. We have addressed all the comments below. The following major changes will be included in the revision. We hope the responses and planned revisions will be viewed favorably.
>
> 1. The theoretical analysis may be able to be improved. The current proof only shows convergence in limit, maybe can obtain an explicit convergence rate.
> >Yes, we can obtain an explicit convergence rate. By revisiting the proof of Theorem~1, we found that the rate of convergence is at the order of $O(\sum_{\tau=1}^t (a_{1:\tau}/a_{1:t})  a_{\tau}^2)$. We have revised the theorem statement in Appendix B (highlighted in the revision).

---

> > ### Comment · Reviewer_3PDZ · 2022-08-09
> > **Thanks for your clarification**
> >
> > Thank you for clarify my comment. I think this paper should be accepted.

---

> ### Author Response · Authors · 2022-08-06
> **A kind reminder**
>
> Dear Reviewer 3PDZ,
>
> We would like to thank you again for the time you dedicated to reviewing our paper and your valuable comments. We believe that we have addressed your concerns. Since  the end of discussion period is getting close and we have not heard back from you yet, we would appreciate if you kindly let us know of any other concerns you may have, and if we can be of any further assistance in clarifying any other issues.
>
> Thanks a lot again, and with sincerest best wishes
>
> Authors

---

> ### Author Response · Authors · 2022-08-09
> **A kind reminder**
>
> Dear Reviewer 3PDZ,
>
> We apologize for any inconvenience that our message may cause in advance. Again, we would like to thank you for the time you dedicated to reviewing our paper and your valuable comments. We believe that we have addressed your concerns. Since the end of discussion period is close and we have not heard back from you yet, we would appreciate if you kindly let us know of any concerns you may have, and if we can be of any further assistance in clarifying any other issues. We humbly remain at your disposal.
>
> Thanks a lot again, and with best wishes,
>
> Authors

---

### Meta-Review · Area_Chair_R4qX · 2022-08-23

**Recommendation:** Accept
**Confidence:** Certain

**Metareview:**

This paper proposes and analyzes a new approach, Gradient Assisted Learning, to collaborative training when features are split across several organizations. Although the work has some limitations, all reviewers agreed that the paper contains a strong, sound contribution and should be accepted. The revisions made during the post-rebuttal discussion already address the majority of the concerns raised, which were about presentation of the results and clarifying some points wrt related work.

**Award:**

No

---

### Decision · Program_Chairs · 2022-09-14

Accept